# Associations between the Mediterranean Diet Pattern and Weight Status and Cognitive Development in Preschool Children

**DOI:** 10.3390/nu13113723

**Published:** 2021-10-22

**Authors:** Federico Granziera, Maria Angela Guzzardi, Patricia Iozzo

**Affiliations:** 1Institute of Clinical Physiology, National Research Council (CNR), 56124 Pisa, Italy; federico.granziera@santannapisa.it (F.G.); patricia.iozzo@ifc.cnr.it (P.I.); 2Sant’Anna School of Advanced Studies, 56127 Pisa, Italy

**Keywords:** early childhood, nutrition, Mediterranean diet, body mass index, cognitive development

## Abstract

Cognitive dysfunctions are a global health concern. Early-life diet and weight status may contribute to children’s cognitive development. For this reason, we explored the associations between habitual food consumption, body mass index (BMI) and cognitive outcomes in 54 preschool children belonging to the Pisa birth Cohort (PISAC). We estimated groups of foods, nutrients and calorie intakes through a food frequency questionnaire (FFQ) and Italian national databases. Then, we adopted the Mediterranean diet (MD) score to assess relative MD adherence. Cognition was examined using the Griffiths Mental Development Scales-Extended Revised (GMDS-ER). We found that higher, compared to low and moderate, adherence to MD was associated with higher performance scores. Furthermore, white meat consumption was positively related to BMI, and BMI (age–gender specific, z-scores) categories were negatively related to practical reasoning scores. All associations were independent of maternal IQ estimates, parents’ socioeconomic status, exclusive/non-exclusive breastfeeding, actual age at cognitive assessment and gender. In conclusion, in preschool children, very high adherence to MD seemed protective, whereas BMI (reinforced by the intake of white meat) was negatively associated with cognition.

## 1. Introduction

Cognitive decline and impairment are globally increasing health concerns, associating with a growing prevalence of metabolic diseases [1] and with population aging [2]. Effective treatment is lacking, and early prevention targeting modifiable determinants is warranted [3]. Life-course studies have shown that a lower intelligence quotient (IQ) at 11 years is already predictive of dementia seven decades later [4], suggesting that the risk is partially settled at 11 years and preventive actions should focus on younger children.

Diet and body weight are accredited lifestyle determinants of cognitive (dys)function in adults and patients [5], but few studies have explored associations between nutrients or consumption of given foods or eating habits and cognitive outcomes in children from developed countries. Some have investigated the effects of individual micronutrients, such as vitamin B12, folic acid, zinc, iron, and iodine, but findings in non-IQ-deficient children are controversial [6]. In addition, studies investigating the association of plasma biomarkers of polyunsaturated fatty acids and cognition in children have provided inconsistent results [7]. For example, a study observed a direct relationship between the proportion of docosahexaenoic acid (DHA) and eicosapentaenoic acid (EPA) in blood and working memory in children [8]; instead, Boucher et al. found no associations between the proportions of DHA, EPA, or other omega-3 fatty acids in blood and working memory in children [9]. Several other studies have focused on macronutrients and whole foods [10]. For example, an observational study in 586 European children aged 7–9 years documented that consumption of two fish (including one fatty fish) meals per week reduced social, attention, and behavioural problems [11]. Another study in 5200 Canadian children aged 10–11 showed that lower fat and higher fruit and vegetable intakes were associated with better reading and writing achievements [12]. Sugar intake did not emerge as a factor affecting behaviour or cognitive performance in children in meta-analysis studies [13], but consumption of sugar-sweetened beverages was recently reported as negative predictor of higher verbal scores in 3-year-old American children [14]. Other studies found that low-glycaemic index (GI) breakfasts predicted better attention and memory, and declarative-verbal memory and high-GI breakfasts were associated with better vigilance in 6–11 and 11–14-year-old children [15,16]. Taki et al. showed that brain grey and white matter volumes were greater in children eating rice than bread at breakfast, speculating that this may depend on the lower glycaemic index of rice [17]. However, a recent systematic review concluded that there is a lack of research comparing breakfast types, precluding recommendations for the size and composition of an optimal breakfast for children’s cognitive function [18].

Dietary patterns and diet quality indices have been suggested to better reflect a real-life diet, where foods are the combination of various nutrients that act synergistically and are interrelated [19,20]. These diet quality indices mostly describe a diet high in vegetables, fruit and berries, non-refined cereals products and fish and low in meats and saturated fat. Among whole-diet scores, the Healthy Eating Index (HEI-2005), Dietary Approaches to Stop Hypertension (DASH) score, Baltic Sea Diet Score (BSDS) and the Finnish Children Healthy Eating Index (FCHEI) were linked to better reading skills or cognition among 6–9-year-old children [7].

This overview highlights that the current understanding of the diet–cognitive development relationship is limited, and a great majority of studies have been focused on children who have entered school, in whom the influence of academic education may represent a relevant confounder, whereas very sparse knowledge has been produced in preschool children.

The aim of this study was to explore, comparatively, the impact of food habits, amount of ingested daily calories, macro- and micronutrient intakes, and Mediterranean diet (MD) adherence in preschool children on cognitive outcomes while taking into account the most important known confounders, i.e., maternal IQ estimate, parental socio-economic status, exclusive or non-exclusive breastfeeding, actual age at cognitive assessment and children’s gender.

## 2. Materials and Methods

### 2.1. Study Population

The study was conducted in a subgroup of *n* = 54, 5-year-old preschool children of the Pisa birth Cohort (PISAC). Overall, the PISAC cohort includes 90 families—father, mother and infants born in 2011–2014 and living in Tuscany, Italy—enrolled during pregnancy to investigate the effects of maternal obesity on offspring cardiometabolic and cognitive health. The cohort was intended to represent the general population and, therefore, the inclusion criteria were broad, namely, (1) mothers within the first trimester of pregnancy at the first visit or at delivery; (2) any parents’ age; (3) any BMI; (4) willingness of mothers and fathers to participate and to actively collect questionnaires and samples; (5) capacity of mothers and fathers to understand the study and its implications; (6) signature of the informed consent by mothers and fathers; (7) absence of major diseases (mothers and fathers) and perinatal complications. Exclusion criteria were (1) history of major diseases in the mother and in the father (kidney failure, liver failure, cardiac failure, major lung disease, autoimmune disease, cancer, psychiatric illness, also including anorexia–bulimia nervosa and substance abuse); (2) major health complications during the perinatal period; (3) failure to understand the study’s implications, comply with the study’s schedule or sign the consent form. Follow-up visits were carried out from birth to the children’s age of five years and consisted of anthropometric, echocardiographic and cognitive assessments and collection of biological samples (cord–blood, faeces and saliva) and of FFQs (at 5 years). Families were given the option to participate in all or only part of the assessment visits (0, 12, 18, 24, 36 and 60 months of life); therefore, the sample size varied across age points and measurements. In particular, families not included in this analysis did not have time to attend all 60 months’ visits and chose to have cardiac or anthropometric evaluations rather than cognitive assessments or no evaluation at all. At 60 months, children’s body weight (in kg to the nearest 0.1 kg) and length (in cm to the nearest 0.5 cm) were measured by weight scale and stadiometer, with children wearing light clothes and standing straight without shoes and with heels close together [21]. Then, the children’s BMI (kg/m^2^) was calculated, and BMI-for-age (gender-specific, z-scores) categories were defined as follows: moderately underweight (>−3 to <−2 standard deviations), normal weight (>−2 to <+1 standard deviations), overweight (>+1 to <+2 standard deviations), obesity (>+2 standard deviations) [22]. Data on parents’ jobs were transformed in socioeconomic classes using the European Socio-economic Classification (ESeC) [23].

The study was conducted in accordance with the Declaration of Helsinki and approved by the Ethics Committee of Massa and Carrara and the latest amendments by the Ethical Committee of the Area Vasta Nord-Ovest (CEAVNO), Pisa, Italy (Study ID 394, approval decree/document n. 75 and 71512). Parents gave their written informed consent before inclusion.

### 2.2. Food Frequency Questionnaire (FFQ)

Dietary assessment was conducted for 65 children throughout a validated, self-administered, semi-quantitative FFQ, with minor modifications [24]. However, the current analyses pertained to the 54 children who also underwent the cognitive visit. The FFQ consisted of 53 commonly used food items (including 124 foods) classified into 22 groups (bread, pizza, crackers and breadsticks, pasta or rice, minestrone soup with pasta or rice, barley and spelled, polenta, couscous, potatoes, eggs, fresh and processed meats, fish, cheeses, milk, yogurt, vegetables, olives, fruit and nuts, legumes, cakes and snacks, sugar and honey teaspoons added to milk and drinks and beverages). Vegetable drinks, milkshakes, wine and beer were included in the FFQ but were not consumed in our population. The type of fat (extra virgin olive (EVO) oil, olive oil, seed oil, butter, margarine, cooking cream, bacon and lard) used for preparing, cooking and dressing food was also addressed.

Frequency response categories for foods items were the following: never, less than once a month, 1–3 times a month, once a week, 2–4 and 5–6 times a week, once a day and 2–3 and 4–5 times a day. Frequency response categories for cooking and dressing fats included: always (2–3 times per day), sometimes (twice per month) and never. The parents filled in the FFQ on the child’s behalf. Data were checked for completeness and consistency, considering incompleteness in >25% items as grounds for a priori exclusion [25]. All FFQs were valid to be submitted into data processing and analysis. A total of 23 specific food groups were further grouped into 13 broader categories (cereals, potatoes, eggs, red and processed meats, white meat, fish, dairy products, legumes, vegetables, fruit and nuts, cakes and snacks, sugar-sweetened drinks and cooking–dressing fats ratio) based on their nutritional content. To estimate the weekly grams consumed for each food category, we first calculated the consumptions of the 53 food items, multiplying the frequency of consumption by the age-appropriate standard portion [26], and we summed the amounts consumed of each food item belonging to the specific category, thus obtaining relative consumption. To estimate daily nutrients and calories intakes, we started by obtaining the nutritional values of the original 124 foods using the Food Composition Tables for Epidemiological Studies in Italy (BDA) [27] or the Food Composition Tables (Council for Agricultural Research and Analysis of the Agricultural Economy, CREA) [28]. The nutritional content of each food was used to calculate the mean amount/portion for each nutrient in the 53 food items. Then, we obtained the daily intake of each nutrient through the weighted sum of the consumption frequency of each food item by its amount/portion. Furthermore, the daily caloric intake was estimated through the weighted sum of each macronutrient intake by its calories. Among the 65 children, eight participants had food allergies or intolerances (i.e., to gluten, milk or lactose, tomato, chickpeas, sesame, shellfish and egg-white), and eighteen had taken food supplements (i.e., prebiotics, probiotics, vitamins and minerals) in the last month. None of the parents declared any other major food-related illnesses affecting their children.

### 2.3. Mediterranean Diet (MD) Score

The degree of children’s adherence to the traditional Mediterranean diet was estimated according to the score proposed by Trichopoulou et al., with a minor modification [29]. Briefly, based on median consumption values, a score of 0 or 1 was assigned to each of the following 9 food categories: vegetables, legumes, fruits and nuts, cereals, fish, red and processed meats, white meat, dairy products and the ratio of unsaturated fatty acids to saturated fatty acids (cooking–dressing fats). For dietary components that are considered protective in the MD (vegetables, legumes, fruits and nuts, cereal, fish and a high ratio of unsaturated/saturated fatty acids), a score of 0 was attributed if consumption was below the median value of the population, and 1 point was given if it was equal or above the median value. The opposite was done for the other components (i.e., red/processed and white meats and dairy products). Thus, the total MD score ranged from 0 (reflecting no adherence at all) to 9 points (maximal adherence to the traditional MD). Finally, adherence to the MD score was categorised into 4 categories: low (score 0–2), moderate (score 3–4), high (score 5–6), and very high (score 7–9).

### 2.4. Neuropsychological Assessment

The children’s cognitive development was evaluated by a trained psychologist in a dedicated hospital room using the GMDS-ER version [30,31,32] in *n* = 54 of the children, addressing the following 6 cognitive domains: locomotor, personal–social, hearing and language, hand–eye coordination, performance and practical reasoning. Maternal IQ was estimated in *n* = 52 women by the Raven’s progressive matrices.

### 2.5. Statistical Analysis

SPSS for Windows (version 26, Chicago, IL, USA) was used for statistical analysis. Regression models, such as bivariate correlation analysis, were performed to assess associations between continuous variables, and partial correlation analyses to adjust for covariates (i.e., mothers’ IQ estimate, parents’ ESeC, exclusive or non-exclusive breastfeeding, actual age at cognitive assessment and gender). To avoid chance findings, *p*-values were corrected for multiple comparison, using the Benjamini–Hochberg false discovery rate (Q = 0.20). General linear models and *t*-tests or two-way ANOVAs (analysis of variance) were performed for two or more than two group comparisons, and ANCOVA (analysis of covariance) was used to incorporate covariates). The results are presented as the mean ± standard deviation (SD) or standard error (SEM), and *p*-values ≤ 0.05 were established as the threshold for rejecting the null hypothesis.

## 3. Results

### 3.1. Description of the Study Population

The characteristics of the study’s population are reported in Table 1. The number of boys slightly prevailed over that of girls, and nearly half of the children were exclusively breastfed in the first 6 months of life [21]. According to WHO’s child growth standards [22], half of the children were modestly underweight and <20% were overweight. The parents’ mean age was 39.1 ± 4.2 years for mothers and 41.7 ± 4.6 years for fathers. With reference to the ESeC 3-class model [23], parents were well distributed between working and intermediate classes, and were less represented in the salariat class. Finally, children’s mean age at cognitive assessment was 5.2 ± 0.1 and all children were cognitively healthy, like their mothers [31].

### 3.2. Children’s Food, Calories and Nutrients Intake

The consumption of the 13 food categories and nutritional and caloric intake are reported in Table 2. We found no difference between boys and girls, and the average weekly food consumption was mostly in line with the Guidelines for Healthy Italian Food Habits, with the exception of a lower consumption of eggs (<100 g/week) [26]. The estimated calories and (micro-)nutrients amounts were in accordance with the Nutrient and Energy Reference Intake Levels for the Italian population aged 4–6, except for vitamin D, the consumption of which was almost 90% below the recommended amount (10 μg/day) [33]. Among cooking and dressing fats, the intake of unsaturated (over saturated) fats prevailed, reflecting a predominant use of EVO, olive and seed oils.

### 3.3. Correlations between Food Categories and BMI and Cognitive Outcomes

Relevant associations between weekly consumption of foods included in the MD score and children’s BMI or cognitive outcomes are reported in Table 3. Bivariate analysis showed that the consumption of white meat was related to BMI (*p* = 0.005), unsaturated/saturated fats ratio (cooking–dressing) was related to hand–eye coordination (*p* = 0.005), dairy products were related to performance scores and vegetable consumption was associated with personal–social scores (*p* = 0.015). In addition, BMI categories were negatively related to practical reasoning score (*p* = 0.010) (Table 3). The other variables in Table 2 did not show associations with cognitive scores or BMI.

Among potential confounders, we found that some intakes in children were correlated with maternal IQ and parents’ ESeC or exclusive/non-exclusive breastfeeding in the first 6 months of life. In particular, maternal IQ estimate was related to children’s unsaturated/saturated fats ratio intake (r = 0.367, *p* = 0.007), to hand–eye coordination scores (r = 0.305, *p* = 0.035) and practical reasoning scores (r = 0.376, *p* = 0.008); parents’ ESeC was related to children’s cereals intake (r = 0.320, *p* = 0.014) and to unsaturated/saturated fats ratio intake (r = 0.437, *p* = 0.001). Moreover, exclusively breastfed children consumed more potatoes than non-exclusively breastfed children (*t*-test, *p* = 0.017). In addition, the actual age (in months) at cognitive assessment was negatively related with all but two cognitive domains, hearing and language and hand–eye coordination (r = −0.326, *p* = 0.016 locomotor; r = −0.441, *p* = 0.001 personal–social; r = −0.327, *p* = 0.016 performance; r = −0.498, *p* < 0.001 practical reasoning domains). For this reason, the above correlative analyses were adjusted for these variables and for children’s gender. Instead, mothers’ BMI during pregnancy was not related to any dietary or cognitive outcome.

After adjustment, partial correlation analyses showed that white meat consumption was significantly related to BMI, and BMI categories remained negatively related to practical reasoning scores. Instead, associations between food categories and cognitive outcomes did not remain significant (Table 3).

### 3.4. Adherence to MD and Cognitive Outcomes

Continuous data did not show significance. Children were further stratified into tertiles of cognitive scores (gender-specific high, medium and low tertiles), and univariate and adjusted analyses were performed. In the unadjusted model, children with maximum adherence to the MD had higher performance scores than those with low and moderate adherence to the MD (*p* = 0.003, *p* = 0.014, respectively; *p* trend = 0.008). In the other five cognitive domains, no difference was seen between the four MD categories (data not shown). Adjustment for the relevant covariates confirmed the above results, as shown by significance levels given in Figure 1. According to Ivens et al. [31], children in the second tertile were within the average range of performance scores, and children in the third tertile were above average to the very high range. The grouping in tertiles reduced the interindividual variability effects and resulted in the detection of very high MD adherence as a significant thresholding range.

## 4. Discussion

In the present study, we found that stricter MD adherence was positively related with scores in the performance domain in preschool children. Furthermore, children’s BMI was associated with white meat consumption (positively) and with practical reasoning scores (inversely).

Among cognitively healthy school children, two cohort studies have shown that a higher quality diet was associated with better cognitive tasks. Khan et al. explored the association between inhibitory control (Kaufman Brief Intelligence Test or the Woodcock-Johnson Tests of Cognitive Abilities) and overall diet quality (HEI-2005 score) among 65 American 7–9-year-old children using three-day dietary records. They found that the HEI-2005 scores were negatively associated with response accuracy interference, suggesting greater cognitive flexibility [34]. Haapala et al. suggested that diet quality favours precocious non-verbal fluid intelligence and abstract reasoning in 428 Finnish children aged 6–8 years, in whom DASH and BSDS scores, calculated by a four-day food record, were directly associated with Raven’s Coloured Progressive Matrixes scores (i.e., higher cognitive performance) [35]. In the present study, we evaluated diet quality through the use of the MD score. To our knowledge, these findings are the first evidence for the associations of adherence to the MD and cognition in preschool children, even if limited to a small population. The expectation when adopting this score was to reinforce the impact of the single components by their pooling and establish the level of MD adherence that would result into a clinically significant difference. Interestingly, a two-point MD adherence score difference has been shown to lower overall and cardiovascular mortality, and the incidence of Parkinson’s and Alzheimer’s diseases in the general population [29]. Though neural mechanisms underlying cognitive benefits of the MD have not been clarified [36], there is evidence that circulating levels of glucose, choline, tyrosine and tryptophan, polyunsaturated fatty acids, vitamins/minerals, antioxidants and the gut microbiota affect neurochemistry, neurotransmission and neuroprotection in the human brain [37,38,39].

In addition, due to the sample size’s limitation, the ability to control for confounders was limited. Other important confounders may include cognitive stimulation, parental educational attainment, etc.

We aimed to dissect single food categories or macro- and micronutrients that could contribute to explaining cognitive scores in children. However, it may be difficult to separate the specific effects of single nutrients/foods because of the interactive/synergistic nature of nutrients, high intercorrelation among nutrients and foods, potential small effect of a single food or nutrient and the residual confounding by dietary patterns [8]. We applied false discovery rate corrections to our analyses and then adjusted for confounders, maximising rigor and minimising chances for significant findings. Therefore, we consider that bivariate associations surviving false discovery rate corrections retain the value of hypothesis generating observations. They suggest positive effects of high intakes of vegetables (vs. personal–social scores), dairy products (vs. performance scores) and unsaturated/saturated fats ratios (vs. hand–eye coordination scores). These correlations were not significant after introducing parental and children’s confounders, which may be due to the inter-correlation, limited sample and/or lack of a direct relationship and, overall, the correlations must be interpreted in light of the risk of type II error (i.e., false negative).

We noted that the intake of vitamin D in children seemed very low. Though food is not the main source and indicator of circulating vitamin D, there are recommended daily vitamin D intake ranges, and the observation of a 90% reduction below this range in our children is consistent with the estimated 80% prevalence of vitamin D deficiency in children living in developed countries, including countries with ample sunrays [40].

As a secondary study outcome, we found an inverse association between weight status and practical reasoning scores. Childhood overweight and obesity rates have risen dramatically over the past few decades. Although obesity has been linked to poorer neurocognitive functioning in adults, much less is known about this relationship in children and adolescents [41,42]. Our finding is in line with a recent meta-analysis showing a negative relationship between BMI and various aspects of neurocognitive function such as executive function, attention, visuo-spatial performance and motor skills in healthy children [43]. Our results strongly support these authors’ statement, that longitudinal studies are urgently needed to determine the directionality of such relationships and identify the critical intervention time periods in order to develop effective treatment programs [43].

The present findings should be interpreted within the context of the study’s strengths and limitations. Strengths of the present study include the assessment of children’s diet with a validated dietary questionnaire, well-established outcome measures and control for several family, maternal and children’s characteristics. We opted for the FFQ to reflect usual eating habits rather than short-lasting dietary exposures [44], and cognitive development was objectively measured by a trained psychologist by using the GMDS-ER test. Dietary estimations were based on a total of 124 foods using validated Italian food databases; however, given the FFQ format, we recognise that adopting standard portion sizes may be less accurate than actually measuring portions. Another important limitation is the small sample size. Therefore, our findings should be interpreted as hypothesis generating observations. We also acknowledge that there are possibly more comprehensive tests to assess cognition at the age of 5 years, but our children were followed from the age of 6 months, and we chose to preserve the same (though age-adapted) test longitudinally. We recognise that comparison with other studies is rather complex, mainly because of different methodological approaches, including dietary records, type of dietary scores/indices, control of confounding factors, type of cognitive subtests, age range of children, with very few studies addressing preschool children. Although our results incorporated information on the most known relevant confounders, we acknowledge that the ability to control for confounders could be reduced because of the sample size, and that residual confounding effects related to other unmeasured variables, such as cognitive stimulation, parental educational attainment and physical activity, may still occur. Moreover, the age at cognitive assessment had a negative association with cognitive outcomes, but we can only speculate that other unaccounted factors could be involved. Finally, the cross-sectional design cannot establish causal relationships.

## 5. Conclusions

The present study provides limited but novel evidence linking children’s diet, weight status and cognition, suggesting potentially positive impacts of the Mediterranean diet and negative impacts of a high BMI. Our data are compatible with the hypothesis that an optimal intake of key foods, acting in synergy with the MD, may confer protection. Our hypothesis-generating findings on single foods support the conduct of larger prospective and mechanistic studies to confirm the positive relationship observed between intake of dairy products, vegetables or unsaturated (vs saturated) fats and cognitive outcomes.

## Figures and Tables

**Figure 1 nutrients-13-03723-f001:**
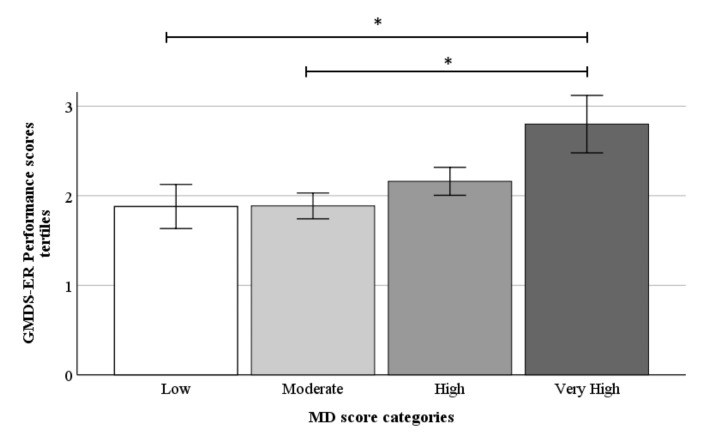
The children’s performance in tertiles (low = 1<; medium = >1 to 2<; high = >2) stratified by the Mediterranean diet score categories (*n* = 44). Data are shown as mean ± SEM. * *p* < 0.05.

**Table 1 nutrients-13-03723-t001:** Characteristics of the study population.

Variable	*N*	Descriptive Results
Boys/girls, *N* (%)	54	30 (55.6)/24 (44.4)
Breastfeeding (exclusively/non-exclusively), *N* (%)	54	25 (46.3)/29 (53.7)
Weight at 5 years (kg), mean ± SD	54	20.6 ± 3.7
BMI at 5 years (kg/m^2^), mean ± SD	54	17.1 ± 2.4
BMI UW/NW/OW/OB at 5 years, *N* (%)	54	27 (50)/18 (33.3)/9 (16.7)/0
BMI UW/NW/OW/OB at 5 years (kg/m^2^), mean ± SD	54	15.3 ± 0.9/17.6 ± 0.5/21.1 ± 0.5/0
Mothers’ BMI in pregnancy	51	29.4 ± 4.9
Mother’s age (years), mean ± SD	54	39.1 ± 4.2
Father’s age (years), mean ± SD	54	41.7 ± 4.6
Mothers’ ESeC WK/IN/SA, *N* (%)	53	18 (33.3)/21 (38.9)/14 (25.9)
Fathers’ ESeC WK/IN/SA, *N* (%)	48	19 (35.2)/19 (35.2)/10 (18.5)
Mothers’ IQ estimate, mean ± SD	48	114.9 ± 9.5
Actual age at cognitive assessment	54	5.2 ± 0.1
Locomotor score, mean ± SD	54	103.8 ± 7.3
Personal–social score, mean ± SD	54	104.8 ± 8.6
Hearing and speech score, mean ± SD	54	101.6 ± 10.0
Hand–eye coordination score, mean ± SD	54	99.2 ± 10.0
Performance score, mean ± SD	54	112.0 ± 7.1
Practical reasoning score, mean ± Practical SD	54	95.7 ± 9.2
MD scores, mean ± SD	54	4.3 ± 1.6
MD score 0–2/3–4/5–6/7–9 categories, *N* (%)	54	8 (14.8)/20 (37)/20 (37)/6 (11.1)

Population characteristics are given as the mean ± SD or number and (%), as appropriate. UW = moderately underweight, NW = normal weight, OW = overweight, OB = obesity, WK = working class, IN = intermediate class, SA = salariat class and MD = Mediterranean diet.

**Table 2 nutrients-13-03723-t002:** Children’s weekly food groups and daily energy and nutrients intakes.

Dietary Variable	*N*	Children’s Intake
Cereals (g/week)	53	1110.8 ± 484.0
Potatoes (g/week)	53	173.1 ± 124.9
Legumes (g/week)	53	41.5 ± 38.9
Eggs (g/week)	53	49.5 ± 47.4
Red and processed meats (g/week)	54	140.6 ± 96.6
White meat (g/week)	54	120.8 ± 70.1
Fish (g/week)	54	137.5 ± 104.0
Dairy products (g/week)	54	582.3 ± 486.5
Vegetables (g/week)	54	492.5 ± 447.1
Fruit and nuts (g/week)	54	1152.5 ± 868.8
Cakes and snacks (g/week)	54	406.8 ± 358.7
Sugar-sweetened drinks	53	865.5 ± 644.3
Unsaturated/saturated fats ratio	54	4.7/1 ± 1.1
Daily calorie (kcal/day)	54	1569.0 ± 394.7
Proteins (g/day)	54	48.3 ± 13.4
Lipids (g/day)	54	60.4 ± 13.9
Carbohydrates (g/day)	54	202.4 ± 61.5
Fibres (g/day)	54	11.0 ± 3.7
Retinol (mg/day)	54	414.1 ± 163.6
Vitamin B1 (mg/day)	54	0.5 ± 0.1
Vitamin B6 (mg/day)	54	0.9 ± 0.2
Folate (μg/day)	54	144.4 ± 48.2
Vitamin C (mg/day)	54	75.5 ± 48.7
Vitamin D (μg/day)	54	0.6 ± 0.3
Vitamin E (mg/day)	54	7.9 ± 1.6
Iron (mg/day)	54	5.2 ± 1.5
Calcium (mg/day)	54	649.1 ± 245.0
Sodium (mg/day)	54	1264.0 ± 468.6
Potassium (mg/day)	54	1728.2 ± 481.7
Phosphorous (mg/day)	54	797.8 ± 241.4
Zinc (mg/day)	54	5.4 ± 1.5

Continuous data are reported as the mean ± SD.

**Table 3 nutrients-13-03723-t003:** Correlations between MD food categories and BMI (including BMI categories) and cognitive outcomes.

Variables	Regression Model	BMI	Locomotor	Personal–Social	Hearing and Language	Hand–Eye Coordination	Performance	Practical Reasoning
Cereals (g/week)	Bivariate	−0.188	0.029	0.208	−0.010	0.032	−0.030	0.137
	Adjusted	−0.042	0.004	0.274	0.109	0.035	−0.055	0.227
Potatoes (g/week)	Bivariate	−0.077	0.078	−0.012	−0.021	0.037	−0.057	0.116
	Adjusted	−0.307	0.114	−0.049	−0.210	0.008	−0.010	0.158
Legumes (g/week)	Bivariate	0.096	0.011	0.061	0.002	−0.093	0.103	−0.105
	Adjusted	0.113	0.079	0.104	−0.009	−0.044	0.188	−0.146
Eggs (g/week)	Bivariate	−0.162	−0.085	0.068	−0.130	0.229	0.188	−0.028
	Adjusted	−0.059	−0.089	−0.049	−0.213	0.380	0.199	−0.113
Red and processed meats (g/week)	Bivariate	0.237	0.033	0.097	0.121	0.012	−0.007	−0.006
	Adjusted	0.319	0.185	0.217	0.188	0.059	0.057	0.056
White meat (g/week)	Bivariate	0.377 **	0.081	0.084	0.054	0.009	−0.111	−0.080
	Adjusted	0.440 **	0.267	0.303	0.067	0.089	0.023	−0.015
Vegetables (g/week)	Bivariate	−0.067	0.242	0.295 *	0.067	0.037	0.101	0.030
	Adjusted	0.073	0.285	0.315	−0.009	0.051	0.005	−0.006
Fruit and nuts (g/week)	Bivariate	−0.030	0.059	0.044	−0.168	0.018	−0.124	−0.032
	Adjusted	0.085	0.053	0.046	−0.088	0.199	−0.102	0.181
Dairy products (g/week)	Bivariate	−0.232	0.030	−0.015	0.053	0.101	0.275 *	0.233
	Adjusted	−0.177	−0.021	−0.093	0.024	0.027	0.272	0.233
Unsaturated/saturated fats ratio	Bivariate	−0.023	−0.078	−0.018	−0.037	0.369 **	0.254	0.212
	Adjusted	−0.009	−0.282	−0.224	−0.086	0.295	0.222	−0.002
BMI categories	Bivariate	0.792 **	−0.131	−0.083	−0.115	−0.005	−0.100	−0.358 **
	Adjusted	0.811 **	−0.122	0.084	−0.126	0.100	0.063	−0.329 *

The table provides the results of the regression analysis for both the bivariate and adjusted models. Bivariate analyses were performed in *n* = 53 (list-wise deletion) and partial correlation analyses were performed in *n* = 43 (list-wise deletion); *p*-values were corrected for the false discovery rate using the Benjamini–Hochberg (Q = 0.20). * *p* < 0.05, ** *p* < 0.01. Adjustment for confounders in multivariate analyses includes mothers’ IQ estimate, parents’ ESeC, exclusive/non-exclusive breastfeeding, actual age at cognitive assessment and gender.

## Data Availability

The data presented in this study are available on request from the corresponding author.

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
