# Peer review of "Associations between the Mediterranean Diet Pattern and Weight Status and Cognitive Development in Preschool Children"

_nutrients, 2021, doi:10.3390/nu13113723_

Round 1

Reviewer 1 Report

There are a few well known methodological and conceptual limitations in assessing diseases in relation to dietary intake as described by Hu (2002). The study of single nutrients or foods can be complicated by the interactive/synergistic nature of nutrients, high intercorrelation among nutrients and foods, potential small effect of a single nutrient, chance finding of significant association due to large number of nutrients/foods being assessed, and residual confounding by dietary patterns. Dietary index approach is limited by current understanding of diet-disease relationship and the uncertainties in selecting individual components of the score and subjectivity in defining cut-off points. It is unclear to me how the manuscript built on the understanding of these issues.     

Another major concern is the small sample size. Although 65 children were included in the cohort, only 54 had cognitive assessment, and a few covariates appear to have substantial missing data (e.g., mother’s IQ was missing for 20% of the sample). If missing data were handled with list-wise deletion (which should have been made clear), the actual analytic sample must be smaller than 52. As a result, the multivariable models could be overfitting. If parent’s ESeC were specified as multiple dummy variables, 9 variables were included in the multivariable analyses, which was way more than what may be feasible. As a rule of thumb at least 10 observations per predictor variable is needed for reliable estimations (Harrell 2015).

Related to the above comment, instead of showing the characteristics of the analytic sample that was used to estimate the association between the dietary variables and cognitive outcomes, the authors presented the descriptive statistics of the maximum available individuals. Moreover, SEM (which the authors never provided the full form) should not be used to express variability of data, because it is an inferential parameter, quantifying uncertainty in the estimate of the mean. Instead, standard deviations should be reported, which will indicate almost certainly larger variability than the SEMs suggest.

The authors used adherence to MD score as dependent variable and GMDS score tertiles as independent variables (results 3.4). This is conceptually confusing, should not the causal direction go the other way?! Figure 1 shows almost an identical predicted MD scores between groups of medium and high tertiles of GMDS performance, and yet the difference was statistically significant?!   

Given the wide range of available nutrients and foods, why Table 3 presented only the associations of 5 foods and 2 nutrients? Were p-values corrected for multiple comparison? If not, the risk of chance findings could be high because of the large number of comparisons (80 coefficients were reported in Table 3).  

When were dietary intake and cognitive ability were assessed? Were all children aged precisely 5 years at assessment? For preschool children even a few months difference in age could make a big difference in cognition and anthropometric measures. Age (in months) at survey should be included as a covariate given the rapid growth and development of the body.   

A few more detailed comments:

The relationship between metabolic conditions and cognitive impairment is not unambiguous. The book chapter of Citation 1 concluded that “Despite many studies documenting the association between diabetes and cognitive impairment, the causal nature and the mechanisms of this association have not been fully established and require further study.” Is it appropriate to state that cognitive decline and impairment is ‘fostered by a growing prevalence of metabolic diseases?’

Please provide some details about GMDS-ER measure (e.g., a few citations indicating the validity of the measure, the range of the score, and how to interpret the values)?

What was adjusted 1/2 mean in Table 3? Why only the first two food items adjusted 1/2?

How were BMI and BMI categories measured/defined? What were the categories included? Please provide the descriptive statistics in Table 1.  

What is the rationale of controlling BMI in the association of dietary intake and cognitive outcomes? BMI is likely a result of dietary intake (and physical activity) and thus represents a mechanism. Why controlling for an intermediate variable?

Author Response

Review Comments to the Authors:

Reviewer #1:

There are a few well known methodological and conceptual limitations in assessing diseases in relation to dietary intake as described by Hu (2002). The study of single nutrients or foods can be complicated by the interactive/synergistic nature of nutrients, high intercorrelation among nutrients and foods, potential small effect of a single nutrient, chance finding of significant association due to large number of nutrients/foods being assessed, and residual confounding by dietary patterns. Dietary index approach is limited by current understanding of diet-disease relationship and the uncertainties in selecting individual components of the score and subjectivity in defining cut-off points. It is unclear to me how the manuscript built on the understanding of these issues.  

Thank you for the feedback, and for the many useful suggestions and references. There are certainly major limitations in assessing diseases in relation to diet, and diet-diseases relationships were beyond the scope of the present study. Our primary intention was to identify specific and coherent dietary patterns and categories of foods, and secondly nutrients, relating to cognitive development. In fact, our children’s population was cognitively healthy and their cognitive scores were spread over the normal range. The intent was to provide hypothesis generating findings. We have extensively revised our manuscript to address the points of ambiguity (in the introduction, discussion and conclusions), as detailed below, also using the above words of the reviewer in the discussion.

Another major concern is the small sample size. Although 65 children were included in the cohort, only 54 had cognitive assessment, and a few covariates appear to have substantial missing data (e.g., mother’s IQ was missing for 20% of the sample). If missing data were handled with list-wise deletion (which should have been made clear), the actual analytic sample must be smaller than 52. As a result, the multivariable models could be overfitting. If parent’s ESeC were specified as multiple dummy variables, 9 variables were included in the multivariable analyses, which was way more than what may be feasible. As a rule of thumb at least 10 observations per predictor variable is needed for reliable estimations (Harrell 2015).

We recognize that the small sample size is a major limitation of the present study, and in the revised version of the manuscript, we have emphasized this limitation at various levels throughout the discussion (lines 285-287, 333-334) and conclusions (lines 349-352). Also, we presented descriptive results of the cohort characteristics limited to the analytic sample (n=54) (Table 1, Table 2). In addition, we declared that missing data were handled with list-wise deletion, specifying for each analysis the number of children included. As suggested by the Reviewer, the multivariate models could be overfitted, so we considered a more suitable number of confounders, including the actual age at cognitive assessment, for a total of 6 variables included in the multivariate analyses.

Related to the above comment, instead of showing the characteristics of the analytic sample that was used to estimate the association between the dietary variables and cognitive outcomes, the authors presented the descriptive statistics of the maximum available individuals. Moreover, SEM (which the authors never provided the full form) should not be used to express variability of data, because it is an inferential parameter, quantifying uncertainty in the estimate of the mean. Instead, standard deviations should be reported, which will indicate almost certainly larger variability than the SEMs suggest.

Thank you for the precious indication. In the revised version, we present descriptive results as mean ± SD. Also, all abbreviations in the text have been explained.

The authors used adherence to MD score as dependent variable and GMDS score tertiles as independent variables (results 3.4). This is conceptually confusing, should not the causal direction go the other way?! Figure 1 shows almost an identical predicted MD scores between groups of medium and high tertiles of GMDS performance, and yet the difference was statistically significant?!

We entirely agree, and we re-analysed the data, comparing MD score (independent variable) with GMDS score (dependent variable), as suggested. Also, we thank the reviewer for pointing out that Figure 1 was incorrect; in fact, Figures 1a and Figure 1b were erroneously the same. The revised Figure 1 provides the new results, as obtained from the above analysis.

Given the wide range of available nutrients and foods, why Table 3 presented only the associations of 5 foods and 2 nutrients? Were p-values corrected for multiple comparison? If not, the risk of chance findings could be high because of the large number of comparisons (80 coefficients were reported in Table 3). 

In the previous version, in Table 3 were reported dietary variables that had at least a significant result, excluding others non-significant results. In the current version, Table 3 shows all of the tested variables of the MD score, beyond BMI (whether significant or not). The remaining nutrients listed in Table 2 but not in Table 3 were also tested and not significant. To minimize chance findings, p-values were corrected for multiple comparison, using the Benjamini-Hochberg false discovery rate (Q=0.20).

When were dietary intake and cognitive ability were assessed? Were all children aged precisely 5 years at assessment? For preschool children even a few months difference in age could make a big difference in cognition and anthropometric measures. Age (in months) at survey should be included as a covariate given the rapid growth and development of the body.

Thank you for this observation. The actual age at cognitive assessment was included in Table 1, where mean children’s age was 5.2 ± 0.1 years, and age was treated as confounder in the multivariate model.

The relationship between metabolic conditions and cognitive impairment is not unambiguous. The book chapter of Citation 1 concluded that “Despite many studies documenting the association between diabetes and cognitive impairment, the causal nature and the mechanisms of this association have not been fully established and require further study.” Is it appropriate to state that cognitive decline and impairment is ‘fostered by a growing prevalence of metabolic diseases?’

This comment is well taken and we have replaced the term “fostered by” with “associating with” (lines 26-27).

Please provide some details about GMDS-ER measure (e.g., a few citations indicating the validity of the measure, the range of the score, and how to interpret the values)?

On lines 173-176 we added 3 citations indicating in detail the validity of GMDS-ER measure, the range of the score and how to interpret the values.

What was adjusted 1/2 mean in Table 3? Why only the first two food items adjusted 1/2?

We apologize for the lack of clarity in Table 3. Model1 differed from Model2 in the absence of BMI as a confounding variable. However, following the suggestion of the reviewer, this revised version does not utilize the BMI as confounder, and therefore there is only one Model.

How were BMI and BMI categories measured/defined? What were the categories included? Please provide the descriptive statistics in Table 1. 

On lines 103-110 we provided all details about how BMI was calculated and how BMI categories were defined. Also, in Table 1 we provide the descriptive statistics of both BMI (continuous variable) and BMI categories.

What is the rationale of controlling BMI in the association of dietary intake and cognitive outcomes? BMI is likely a result of dietary intake (and physical activity) and thus represents a mechanism. Why controlling for an intermediate variable?

We agree with the reviewer that the rationale of controlling for BMI may be confusing, and therefore BMI was no longer considered as a confounder in the multivariate model.

Reviewer 2 Report

Overall comments

The link between diet, body mass and brain development is an important question to address.

The sample in this study is negligible for an observational cohort attempting to explore this issue, and results are vastly overstated.

Introduction

Introduction is biased towards studies with significant findings. For example, where a trial is compared to the conflicting results of an observational study there was no awareness of the significant differences in these research designs. Additionally, there are numerous meta-analyses of trials in this area that would be better references.

In the rationale much is made about the importance of preschool versus school, yet age of children in this study are considered school age in many countries.

Methods

Inclusion and is exclusion criteria need to be included-all details necessary to appropriately evaluate the sample, without having to view the publications with the original study details.

How many from the original cohort were not eligible for the present analyses, and why.

Given these are preschool children, specify that parents? Completed the FFQ on child’s behalf

How was caloric and nutritional intake calculated? This would be really poor data inaccurate given the format of the FFQ.

Perhaps include more

Why wasn’t maternal BMI taken into consideration? Particularly as this was of primary interest in the enrolment for the original cohort.

Why was the GMDS selected? At 5 years there are several more robust IQ tests available, whereas rhe GMDS is a more basic developmental assessment most commonly used in very young infants/toddlers or young children with severe developmental delays.

Ravens is commonly used as brief inidicator of parental IQ but is not an IQ test, and results can only be reported and discussed as “indicator/estimate of IQ” not “IQ”

It is noteworthy that mean maternal “IQ” as reported is abnormally high

Results

Mean BMI was actually in the underweight range.

Table 1 rows do not align between columns

Results for diet are presented for all 64/65 children, but developmental tests only available for 55.

All results should be based only on the 55 included in the developmental assessments.

Table 2

Unsat/sat fat ratio is not presented as a ratio

“Subjects” should be replaced with “children” or “participants” throughout the manuscript

Discussion

Results are repeated from the results section and throughout the discussion

Results are also overstated, indicating a lack of awareness of the serious limitations of the study-a main one being the incredibly small sample and limited power

The discussion should be abbreviated, removing the repetition and being more succinct.

The conclusion needs to note the limitations

Vitamin D -Main source is the sun, which is not discussed or controlled for. Highly questionable as to whether authors should even have analysed this. Would you expect there to be an effect of dietary vitamin d intake regardless of sun exposure?

Author Response

Review Comments to the Authors:

Reviewer #2:

Overall comments

The link between diet, body mass and brain development is an important question to address.

The sample in this study is negligible for an observational cohort attempting to explore this issue, and results are vastly overstated.

Thank you for the precious feedback and for the important criticism, which we have carefully considered throughout the extensively revised data analysis, discussion and conclusions, as detailed below.

Introduction

Introduction is biased towards studies with significant findings. For example, where a trial is compared to the conflicting results of an observational study there was no awareness of the significant differences in these research designs. Additionally, there are numerous meta-analyses of trials in this area that would be better references.

In the rationale much is made about the importance of preschool versus school, yet age of children in this study are considered school age in many countries.

We have carefully addressed this bias, balancing the introduction between studies with significant and non-significant results that link the child's diet to cognitive outcomes, based on recent meta-analyzes of observational studies and in light of a poor understanding and knowledge of the issue, which remains more than controversial. However, we wished to underline the fact that there is a substantial difference between most published studies in school children and our study, in which all children had not yet entered formal education, a variable that can certainly have a great impact on cognitive development, regardless of actual age. In Italy, primary school is started at 6 years, in other countries it may start earlier, therefore the use of the term preschool-age may be misleading, and we have replaced it with preschool children, eliminating the “age” factor.

Methods

Inclusion and is exclusion criteria need to be included-all details necessary to appropriately evaluate the sample, without having to view the publications with the original study details.

How many from the original cohort were not eligible for the present analyses, and why.

Given these are preschool children, specify that parents? Completed the FFQ on child’s behalf

How was caloric and nutritional intake calculated? This would be really poor data inaccurate given the format of the FFQ.

We thank reviewer for suggestions for clarification. On lines 85-96 we included all details relative to inclusion and exclusion criteria, in order to facilitate their identification and properly evaluate the sample, without referring to our previous study. As reported on lines 98-103, families were given the opportunity to participate in all or only part of the assessments and time points (0, 12, 18, 24, 36, 60 months of child’s life). Parents are busy with work and family, and it may be difficult for them to spend time in the protocol in some periods. Therefore, the sample size varied across follow-up visits. In particular, families not included in this analysis did not have time to attend all 60 months visits and chose to have cardiac or anthropometric evaluations rather than cognitive assessments, or no evaluation at all. On lines 133-134 we declared that parents filled in the FFQ on child’s behalf. On lines 140-152 we added how food categories, nutrients and calories intakes were estimated, and on lines 332-333 (discussion) we stated possible FFQ format limits.

Perhaps include more

Why wasn’t maternal BMI taken into consideration? Particularly as this was of primary interest in the enrolment for the original cohort.

Why was the GMDS selected? At 5 years there are several more robust IQ tests available, whereas rhe GMDS is a more basic developmental assessment most commonly used in very young infants/toddlers or young children with severe developmental delays.

Ravens is commonly used as brief inidicator of parental IQ but is not an IQ test, and results can only be reported and discussed as “indicator/estimate of IQ” not “IQ”

It is noteworthy that mean maternal “IQ” as reported is abnormally high

We thank reviewer for the well taken observations. Maternal BMI was included in the descriptive analyses of the analytical sample (Table 1), providing mean and standard deviation. However, it was not significantly related to any cognitive or dietary variable in these children (results, lines 242-243), and for this reason this variable was not included as a covariate in the multivariate model. As now mentioned in the discussion (lines 335-337), we acknowledge that there are possibly more comprehensive tests to assess cognition at the age of 5 years, but our children were followed from the age of 6 months, and we chose to preserve the same (though age-adapted) test longitudinally. A trained psychologist recommended and performed the GMDS-ER test. Considering that children also undergo other evaluations (cardiac, etc) each visit is quite demanding, and families are more and more busy as children grow, so we preferred not to expand the number of tests. To provide readers with more knowledge on GMDS, on lines 173-176 we added 3 citations indicating in detail the validity of GMDS-ER measure, its age-specificity, the range of the score and how to interpret the values. In addition, we followed the reviewer’s suggestion to replace “mother’s IQ” with “indicator of mother’s IQ” or “mother’s IQ estimate”.

Results

Mean BMI was actually in the underweight range.

Table 1 rows do not align between columns

Results for diet are presented for all 64/65 children, but developmental tests only available for 55.

All results should be based only on the 55 included in the developmental assessments.

Table 2

Unsat/sat fat ratio is not presented as a ratio

“Subjects” should be replaced with “children” or “participants” throughout the manuscript

We thank reviewer for highlighting these errors. We corrected and clarified BMI issues (lines 106-110, 194-195, Table 1). In addition, Table 1 was revised by aligning rows to columns. Also, we presented descriptive results of the cohort characteristics limited to the analytic sample (n=54) (Table 1, Table 2); and, we specified the number of children included in each statistical analysis (Table 3, Figure 1). We thank the reviewer for noticing that unsat / sat was not presented as a ratio (Table 2), which we promptly corrected. Finally, we replaced the word “subjects” with “children” or “participants” throughout the manuscript.

Discussion

Results are repeated from the results section and throughout the discussion

Results are also overstated, indicating a lack of awareness of the serious limitations of the study-a main one being the incredibly small sample and limited power

The discussion should be abbreviated, removing the repetition and being more succinct.

We kindly thank reviewer for comments and suggestions. We avoided repeating results throughout the discussion, which we also abbreviated in relation to the revised results. Also, we were careful in removing any potential overstatement, and in emphasizing the limitations, including sample size, and use of GMDS or FFQ.

The conclusion needs to note the limitations

We extensively reviewed our conclusions, acknowledging the study limitations.

Vitamin D -Main source is the sun, which is not discussed or controlled for. Highly questionable as to whether authors should even have analysed this. Would you expect there to be an effect of dietary vitamin d intake regardless of sun exposure?

We certainly agree that sun exposure makes a difference in vitamin D metabolism. What we wished to underline was that there are recommended intake ranges, and our children were far below, and this was a fact in line with the diffused deficit of this vitamin observed in children, which is not solely due to lack of sun exposure. We have kept this concept and hope it makes sense to the reviewer. However, we have removed correlations with cognition considering that endogenous vitamin D was not known, and this introduces too much uncertainty, as correctly pointed out but the reviewer.

Round 2

Reviewer 1 Report

The revised version represents an improvement over the initial submission. I have a few remaining comments.

  1. Again, type II error (false negative) could be a serious concern, given the small sample size. Out of numerous nutrients and food items assessed in the analyses, no association was identified for cognitive outcomes involving locomotor or personal-social aspect, and only one item has a positive association with other cognitive outcomes each. Also, due to sample size limitation, the ability to control for confounders was limited. Other important confounders may include cognitive stimulation, parental educational attainment, etc.
  2. Why age at cognitive assessment had a negative association with cognitive outcome (lines 231-234)? Shouldn’t the association be positive? Was maternal IQ positively associated with children’s cognitive outcome? Please provide an additional table documenting the coefficients for covariates (perhaps just the analyses with MD score category with performance score) for the reviewing purposes.
  3. In Table 3, why use continuous specification of BMI category as independent variable? Why not use the BMI measure in its original scale (perhaps adding a squared term to model nonlinearity), so that it is easier to interpret the association? Or, if the authors prefer categorical specification, why not show the coefficients associated with underweight, overweight as compared with normal weight?
  4. Figure 1 now makes more sense, but why use performance score tertiles as dependent variable, instead of raw performance score? How to interpret mean performance score tertiles?
  5. Given adherence to MD is one of the key exposure variables, please report the descriptive statistics in Table 1 (N/%).  
  6. It could be interesting to stratify Table 2 by tertiles of overall cognitive outcome, or maybe above and below median.

Author Response

Review Comments to the Authors:

Reviewer #1:

The revised version represents an improvement over the initial submission. I have a few remaining comments.

Thanks for the feedback and for the helpful comments and suggestions. We dedicated to revising the manuscript by addressing all the points raised, and clarifying all points of ambiguity, as detailed below.

  1. Again, type II error (false negative) could be a serious concern, given the small sample size. Out of numerous nutrients and food items assessed in the analyses, no association was identified for cognitive outcomes involving locomotor or personal-social aspect, and only one item has a positive association with other cognitive outcomes each. Also, due to sample size limitation, the ability to control for confounders was limited. Other important confounders may include cognitive stimulation, parental educational attainment, etc.

We agree with the reviewer that given the sample size type II error was a difficult to limit as well as controlling for confounders. On lines 313-316 we stressed that the results obtained from the correlation analyzes must be interpreted in the light of type II error, small sample size and ability to control the confounders. The other suggested confounding factors are appropriate but since we do not have these data, on lines 348-352 we have pointed out this limit.

  1. Why age at cognitive assessment had a negative association with cognitive outcome (lines 231-234)? Shouldn’t the association be positive? Was maternal IQ positively associated with children’s cognitive outcome? Please provide an additional table documenting the coefficients for covariates (perhaps just the analyses with MD score category with performance score) for the reviewing purposes.

We agree that the inverse association between age at cognitive assessment and cognitive outcomes is relevant, but we can only speculate that other factors not taken into account could mitigate this relation (lines 352-354). Mothers’ IQ was related to eye-and-hand coordination and practical reasoning domains (lines 232-233). We evaluated the coefficients for covariates, Eta partial square and p-value, being parameters that allow to evaluate size effect in ANCOVA models. In light of Fig.1, the covariates coefficients suggest that it is appropriate to consider all covariates in our analysis. Here attached please find the file required.

  1. In Table 3, why use continuous specification of BMI category as independent variable? Why not use the BMI measure in its original scale (perhaps adding a squared term to model nonlinearity), so that it is easier to interpret the association? Or, if the authors prefer categorical specification, why not show the coefficients associated with underweight, overweight as compared with normal weight?

We found it appropriate to use BMI-for-age categories than BMI, because at the age of 5 it is of great relevance to account for sex, as pointed out by the WHO. Also, BMI categories allowed to better indicate the related health risks. In addition, we did not show significant results relative to BMI categories/practical reasoning scores from t-test/ANOVA or ANCOVA because bivariate and partial correlation analyses led to similar conclusions. If the reviewer so requests, we can provide additional material for reviewing purposes.

  1. Figure 1 now makes more sense, but why use performance score tertiles as dependent variable, instead of raw performance score? How to interpret mean performance score tertiles?

We appreciate these questions, which helped clarifying the finding. On lines 257-260 we clarified that continuous values did not show significant differences. However, stratifying children’s cognitive scores by tertiles led to significant differences. On lines 265-268 we explained that according to Ivens et al. [31] children from the second tertile were within the average score range, and children in the third tertile were between above average up to very high scores range. The grouping in tertiles reduces interindividual variability effects, and resulted in the detection of very high MD adherence as a significant thresholding range.

  1. Given adherence to MD is one of the key exposure variables, please report the descriptive statistics in Table 1 (N/%).

As request by reviewer, in Table 1 we reported the descriptive statistics for both MD score (mean ± SD) and MD score categories (N/%).

  1. It could be interesting to stratify Table 2 by tertiles of overall cognitive outcome, or maybe above and below median.

We thank the reviewer for the suggestion, we believe it appropriate to report the descriptive statistics of food consumption in Table 2 in order to provide the reader with a general analysis of the population's diet. In light of the results of Figure 1, stratifying on the basis of cognitive outcomes would further signal a growing trend of adherence to the Mediterranean diet, and would not provide further points of reflection/comparison, which the current analysis would allow. But if the reviewer meant by cognitive outcome the GMDS-ER total score (which was not in analysis), we believe that describing the population on the basis of this variable may be misleading because, being the total score is given by the arithmetic mean of the 6 cognitive scores, and therefore is only a gross indicator of global development, and not true cognitive domain.